# Mapping Bovine Tuberculosis in Colombia, 2001–2019

**DOI:** 10.3390/vetsci11050220

**Published:** 2024-05-15

**Authors:** D. Katterine Bonilla-Aldana, S. Daniela Jiménez-Diaz, Carlos Lozada-Riascos, Kenneth Silva-Cajaleon, Alfonso J. Rodríguez-Morales

**Affiliations:** 1Research Unit, Universidad Continental, Huancayo 12001, Peru; 2Grupo Colaborativo de Investigación en Enfermedades Transmitidas por Vectores, Zoonóticas y Tropicales de Risaralda (GETZ), Pereira, Risaralda 660001, Colombia; saray.jimenez@uam.edu.co; 3Instituto Geográfico Agustin Codazzi, Pereira, Risaralda 660001, Colombia; ingeocar@utp.edu.co; 4Faculty of Environmental Sciences and Health Sciences, Universidad Científica del Sur, Lima 15307, Peru; 100097494@cientifica.edu.pe (K.S.-C.); arodriguezmo@cientifica.edu.pe (A.J.R.-M.); 5Grupo de Investigación Biomedicina, Faculty of Medicine, Fundación Universitaria Autónoma de las Américas-Institución Universitaria Visión de las Américas, Pereira, Risaralda 660003, Colombia; 6Gilbert and Rose-Marie Chagoury School of Medicine, Lebanese American University, Beirut P.O. Box 36-5053, Lebanon

**Keywords:** tuberculosis, infection, cattle, GIS, geographic information systems, *Mycobacterium bovis*, one health

## Abstract

**Simple Summary:**

Bovine tuberculosis (bTB) represents a significant threat as a zoonosis, particularly in areas with notable pastoral economies. bTB can also result in human infections. Despite its importance, Colombia lacks comprehensive analyses of its prevalence, with no publications utilising geographic information systems (GIS) to understand its spread. This study takes a novel approach to fill this gap, characterising the temporal and spatial distribution of bTB in the country from 2001 to 2019 using GIS-based maps. This innovative method is pivotal in comprehending the temporal and spatial dynamics of zoonotic diseases in Colombia, as bTB exemplifies, underscoring its implications for both Human and One Health approaches.

**Abstract:**

Introduction: Bovine tuberculosis is a zoonotic disease of significant impact, particularly in countries where a pastoral economy is predominant. Despite its importance, few studies have analysed the disease’s behaviour in Colombia, and none have developed maps using geographic information systems (GIS) to characterise it; as such, we developed this study to describe the temporal–spatial distribution of bovine tuberculosis in Colombia over a period of 19 years. Methods: A retrospective cross-sectional descriptive study, based on reports by the Colombian Agricultural Institute (ICA), surveillance of tuberculosis on cattle farms in Colombia from 2001 to 2019 was carried out. The data were converted into databases using Microsoft Access 365^®^, and multiple epidemiological maps were generated with the QGIS^®^ version 3.36 software coupled to shape files of all the country’s departments. Results: During the study period, 5273 bovine tuberculosis cases were identified in multiple different departments of Colombia (with a mean of 278 cases/year). Regarding its temporal distribution, the number of cases varied from a maximum of 903 cases (17.12% of the total) in 2015 to a minimum of 0 between 2001 and 2004 and between 2017 and 2019 (between 2005 and 2016, the minimum was 46 cases, 0.87%). Conclusions: GIS are essential for understanding the temporospatial behaviour of zoonotic diseases in Colombia, as is the case for bovine tuberculosis, with its potential implications for the Human and One Health approaches.

## 1. Introduction

Bovine tuberculosis is caused by *Mycobacterium bovis*, a member of the Mycobacteriaceae family, alongside the *Mycobacterium tuberculosis* complex. Its impact extends beyond livestock to humans (a zoonosis), posing significant public health concerns [1]. *Mycobacterium bovis* thrives in various environments, exhibiting resilience against harsh conditions. It persists in certain regions despite control efforts, prompting ongoing research into effective prevention and management strategies [2]. Understanding its transmission dynamics is crucial for devising targeted interventions to curb its spread and minimise its consequences for animal and human populations [3,4].

Primarily transmitted through close contact with infected cattle, it can also affect many other animals. Notably, goats, buffaloes, and dogs are susceptible, alongside less expected hosts such as primates, ferrets, and opossums. Even endangered species like the white rhinoceros are at risk. This pathogen’s ability to infect diverse hosts underscores the importance of vigilant monitoring and control measures [5]. Despite the disease’s name suggesting a primary association with bovines, its potential transmission routes and hosts extend beyond cattle. Consequently, comprehensive surveillance and management strategies are essential to prevent its spread across animal populations and mitigate the risks of human exposure [6]. Combating bovine tuberculosis requires interdisciplinary collaboration, combining veterinary expertise, wildlife management, and public health initiatives. By understanding and addressing the multifaceted nature of this disease, stakeholders can work towards minimising its impact on animal welfare and human health [5,7]. In some cases, these host animals can contract the disease from contact with infected cattle or contaminated environments [8]. Unfortunately, in multiple countries, there is a lack of studies on bovine tuberculosis, as it occurs in many areas of Latin America and the Caribbean, including Colombia.

On the other hand, *Mycobacterium tuberculosis* is the bacterium responsible for causing tuberculosis (TB) in humans [9,10,11]. It is closely related to *Mycobacterium bovis* and shares many similarities regarding disease presentation, transmission, and characteristics [12]. However, *Mycobacterium tuberculosis* is adapted to infect humans and is the primary causative agent of human tuberculosis [13]. *Mycobacterium bovis* is the predominant species affecting livestock in most areas.

Although bovines and various other animals have the potential to contract *Mycobacterium tuberculosis*, such infections are regarded as incidental and do not constitute a substantial component of the bacterium’s inherent life cycle [14]. In the context of bovine tuberculosis, the principal apprehension revolves around the transfer of *Mycobacterium bovis* from cattle to humans, which can occur via the consumption of tainted dairy items or through direct interaction with infected animals [15,16].

Efforts to control bovine tuberculosis are multifaceted and crucial for safeguarding animal welfare and public health. Regular testing of cattle herds is fundamental, enabling the early detection of infected animals and preventing further spread within the herd. In the Colombian context, routine diagnostic screenings fundamentally use the tuberculin test. The culling of infected animals, although a contentious measure, is often deemed necessary to halt transmission and prevent widespread outbreaks [17]. Moreover, strict hygiene practices in livestock management, such as maintaining clean living conditions and preventing contact between infected and healthy animals, are indispensable for curbing the disease’s spread to humans. In humans, tuberculosis remains a formidable global challenge, requiring concerted efforts for adequate control. Early diagnosis through robust screening programs is pivotal in identifying and isolating cases promptly, curbing further transmission. Equally essential is ensuring access to proper treatment regimens, including antibiotic therapy and, in some cases, vaccination [18]. Public health interventions, such as education campaigns and improved healthcare infrastructure, are vital in raising awareness, reducing stigma, and minimising the disease’s impact on vulnerable populations. Through collaborative and comprehensive approaches encompassing both veterinary and human health sectors, significant strides can be made in controlling bovine tuberculosis and mitigating its far-reaching consequences [19,20,21,22].

Different tests for diagnosing this pathology include immunological tests (e.g., intradermic reaction or tuberculin), microscopical detection of the fast-acid bacilli, cultures, and PCR, among others [23]. In Colombia, the most common is the tuberculin test, where the antigen, a mixture of proteins from the bacterium, is injected intradermally [24]. From 48 to 96 h after application, the area is evaluated to identify if there has been a reaction, mainly looking for evidence of an increase in the skin’s thickness, which is called a delayed hypersensitivity reaction.

Sometimes, infected domestic animals or zoos receive antibiotic treatment, but slaughter is carried out in the case of bovines. It is important to remember that tuberculosis is a notifiable disease in Colombia and other countries [25]. Its prevention and control are of great importance; constant disinfection of the facilities and regular review by veterinarians can help prevent the spread of tuberculosis in bovines [26].

Bovine tuberculosis is a significant veterinary and public health concern in many parts of the world, including Latin America. The epidemiology of bovine tuberculosis in Latin America can vary among countries due to differences in livestock practices, veterinary infrastructure, socioeconomic factors, and wildlife reservoirs [27,28]. Bovine tuberculosis is present in several countries across Latin America, with varying prevalence levels. Countries such as Mexico [29], Brazil [30], Argentina [31], Chile [32], and Uruguay [33] have reported cases of bovine tuberculosis [34,35]. Unfortunately, the number of studies published in Colombia about bovine tuberculosis, as well as on other infectious diseases in bovines [36,37,38,39,40,41,42,43], is minimal.

The present study aimed to characterise the spatio-temporal spread of bovine tuberculosis from 2001 to 2019 in Colombia.

## 2. Materials and Methods

### 2.1. Type of Study

A descriptive ecological observational, cross-sectional, retrospective assessment of the incidence of tuberculosis in cattle (bovine) was conducted. The bTB annual incidence rate was estimated for all the Colombian departments. This allowed us to develop epidemiological maps of bTB between 2001 and 2019.

### 2.2. Data Source, GIS-Mapping, and Statistical Analyses

In exploring secondary data sources, the leading resource was the bulletins of the Colombian Agricultural Institute (ICA); these contain sociodemographic features and the disease diagnosis. In addition, the total number of cases and their regional locations were taken, and the geographic characterisation of tuberculosis in bovines began. A case was defined as any bovine suspected of having tuberculosis, with or without clinical findings, that was later confirmed by the intradermoreaction test (tuberculin). As this is a screening, no other tests are routinely performed regardless of the result of tuberculin. It is worth noting that the bulletins do not include individual details of animals or other variables, such as the number of veterinarians performing tests. We extracted all the available data from the bulletins and included them in Excel^®^ spreadsheets to perform descriptive analyses, calculate the incidence of bovine tuberculosis, and develop geographical characterisation. Variables include the number of evaluated animals and herds and the number of tuberculin-positive animals and herds per department per year in Colombia within the study period.

For the corresponding geographical characterisation, the open-access app QGIS^®^3.36 was employed. This software contains preloaded geographical tools; for this research, we used Oeste Bogotá (West Bogotá) (Magna SIRGAS) as the primary reference system (EPSG 21896). The development of maps was carried out (at a scale of 1: 1,365,207) and two types of layers were employed; the first one represented the department level using colours of greater and lesser intensity defined by the layer of the epidemiological data (the second layer), and classified by ranges established using quartiles, which allows the areas with a higher incidence of the disease to be differentiated from those with a lower incidence (according to the technique of quartiles/quintiles of cases/100,000 inhabitants). Data were analysed at two levels: herds and individual animals.

Descriptive statistics were employed using the software Stata IC^®^ version 14. The annual total case number per department and the mean annual case number were estimated. As part of the analyses, trend linear regressions from 2005 to 2016 for the country were calculated with a confidence level of 95% (*p* significant <0.05), estimating the determination coefficient (r^2^). Annual incidence rates were defined as the number of TB cases in bovines divided by the population estimates of bovine animals per 100,000 animals for 2016, 2017, 2018, and 2019, as the census was available for those years.

### 2.3. Population and Sample

Colombia’s total bovine population for 2019 was 27,234,027 [44]. The sample was taken from the census data on the ICA databases, from which registered data were obtained for cases of tuberculosis diagnosis in bovines in Colombia from 2001 to 2019 (https://www.ica.gov.co/) (accessed on 10 January 2022). For the period 2016–2019, the annual number of bovines per department was also obtained. The TB incidence rate was also estimated for that range of years by calculating the number of 100,000 bovines per department per year.

## 3. Results

During the study period, 5273 cases of tuberculosis in bovines were identified in the different departments of Colombia (mean 278 cases/year). Regarding its temporal distribution, the number of cases varied from a maximum of 903 cases (17.12% of the total) in 2015 to a minimum of 0 between 2001 and 2004 and between 2017 and 2019 (between 2005 and 2016, the minimum was 46 cases, 0.87%) (Figure 1). A linear regression analysis from 2005 to 2016 showed a statistically significant reduction in the annual national proportion of positive cases (r^2^ = 0.3627, *p* = 0.0382, F = 5.69, root MSE = 0.37512) (Figure 1).

During the study period, 1286 herds in Colombia’s different departments tested positive for bovine tuberculosis. Regarding its temporal distribution, the number of positive herds varied from a maximum of two hundred and six positives (20.22% of the total) in 2012 to a minimum of two (0.16%) in 2001 and 2019. Between 2005 and 2016, the minimum was sixteen positive herds (1.24%) (Figure 2).

Twenty epidemiological maps (Figure 3 and Figure 4) were generated for the geographical distribution of the proportion of animals positive for bovine tuberculosis per department (Figure 3) and the proportion of herds positive for bovine tuberculosis per department (Figure 4).

Between 2001 and 2006, there were 941 cases of bovine tuberculosis among 68,278 animals with a positivity of 1.4%, but between 2005 and 2006, the cases were geographically focalised in three departments, Cundinamarca, Boyaca, and Nariño (Figure 3). From 2007 to 2012, a wider geographical distribution was observed, affecting multiple departments, but Santander in 2011 (12.8%) and Cauca in 2012 (6.5%) were the most affected (Figure 3).

From 2013 to 2016, Huila was the department that was most affected by bovine tuberculosis (5.7%) (Figure 3). In the last three years (2017–2019), no cases of bovine tuberculosis were observed in animals (Figure 3).

Between 2001 and 2006, a total of 69 herds were positive for bovine tuberculosis among 2061 assessed (3.3%), but in 2004, cases were particularly geographically focalised in four departments, Cundinamarca, Boyaca, Antioquia, and Nariño (Figure 3). From 2007 to 2012, a smaller geographical distribution was observed. In Risaralda in 2010 (1/1), the department with the highest proportion of herds that were positive for bovine tuberculosis (Figure 3).

From 2013 to 2016, different departments had herds that were positive for bovine tuberculosis, especially Boyaca, Huila, and Santander in 2015 and 2016 (Figure 3). In the last three years (2017–2019), the number of assessed herds was relatively low (154), but 67 of them were positive for bovine tuberculosis (43.5%), especially Cundinamarca, Boyaca, Antioquia, Cordoba, Quindio, Atlantico, Guajira, and Sucre in 2017 (Figure 3).

Since 2016, data on the number of bovine cases by department have been available from the census. With those data, the rate of cases of bovine tuberculosis per 100,000 animals was calculated (Table 1). No cases were reported at the individual level in 2017 or 2018. In 2016, the department with the highest rate of bovine tuberculosis was Putumayo (22.8 cases per 100,000 bovines) (Table 1), where 45 cases of bovine tuberculosis were reported among the 48,183 animals that were assessed (0.1%). Caldas, Huila, and Cundinamarca also presented high rates of bovine tuberculosis in 2016 (>10 cases/100,000 animals) (Table 1).

## 4. Discussion

Understanding the epidemiology of bovine tuberculosis (bTB) globally poses several challenges, including the complexity of transmission dynamics, which involve interactions between various factors such as the host, pathogen, and environment. Diagnostic tests for bTB vary in sensitivity and specificity, leading to challenges in accurately detecting and diagnosing infected animals. Many countries, especially in low- and middle-income regions, have limited resources for the surveillance and monitoring of bTB. This can lead to an underreporting of cases and hinder our understanding of the disease distribution and trends [45,46].

In many regions, wildlife species serve as reservoirs for bTB, complicating control efforts. Understanding the role of wildlife in bTB transmission and their interaction with domestic animals is crucial for effective disease management. Globalisation and the international trade of livestock contribute to the spread of bTB across borders. Monitoring and regulating the movement of animals can be challenging, and gaps in surveillance and control measures can facilitate the spread of disease [47,48].

Socioeconomic factors such as poverty, inadequate healthcare infrastructure, and cultural practices can influence the prevalence and spread of bTB. Addressing these factors is essential for comprehensive disease control and prevention efforts [49,50].

Bovine tuberculosis is a zoonotic disease, meaning it can be transmitted between animals and humans. Implementing a One Health approach, which considers the interconnectedness of human, animal, and environmental health, is crucial for effectively addressing bTB at the global level. Addressing these challenges requires collaboration between researchers, policymakers, veterinarians, and other stakeholders and investment in surveillance, research, and control measures [51,52,53].

In the case of Colombia, over almost two decades, more than 5000 cases of bTB were reported in the country, with a mean of 278 cases per year. As such, although a reduction in cases is apparent, this represents a significant burden of animal disease and a potential risk to human health as a zoonotic condition [12,54]. In other countries in the Latin American region, such as Brazil, launching and enhancing their National Programs for the Control and Eradication of Brucellosis and Tuberculosis contributes to reducing the prevalence of bovine brucellosis [54]. The program in Colombia for both diseases still needs to be improved to control these bovine diseases better [55].

The National Program for the Prevention, Control, and Eradication of bTB includes the following sanitary measures: a plan for the promotion and prevention of animal health, sanitary and safety authorisation, certification of free-disease herds, and certification for good livestock practices (Figure 5). Recertification is carried out by employing a tuberculin test one year after the first certification on all animals of regulatory sampling age. If the results are negative, a recertification is granted to the property that will be valid for two years (Figure 5). Farms with two negative tuberculin tests within a four-month interval will be recognised as being free of bTB (Figure 5).

Surveillance in Colombia is carried out actively and passively to control the disease. On farms, the detection of bTB-positive animals is carried out through active surveillance and the tuberculin test. In processing plants authorised by the official health entity, passive surveillance is carried out by inspecting the carcasses of 100% of the animals slaughtered daily. Given the findings, public health officials notify the Colombian Agricultural Institute (ICA) of the presence of lesions compatible with bTB; the ICA veterinarian must attend to it and take the samples that are stipulated within the protocol for the diagnosis of the disease no later than 24 h after receiving the notification. Likewise, 100% of the cattle/buffaloes slaughtered in processing plants are inspected for being positive for the tuberculin test (www.ica.gov.co) (accessed on 20 January 2023).

The owner of an animal that must be slaughtered has the right to receive compensation according to the characteristics of the animal (breed, sex, age, production potential, physiological conditions, and genetic value), equivalent to 60% of its actual value, without exceeding the sum of three (3) current legal monthly minimum wages (for 2023, approximately USD 1000) according to Article 1 of resolution 0043, which was published in 2010 by the Ministry of Agriculture and Rural Development (MADR). In 2023, the ICA disbursed approximately USD 545,000 for compensation after sacrificing 774 animals positive for bovine tuberculosis. In addition to all of that, it is essential to mention that, ideally, the number of tested animals should be increased and all bovine individuals should be tested for tuberculin as part of the surveillance; for example, in 2019, when there was more than 27 million animals, all of them would have been tested.

National control programs for bTB are crucial for several reasons. Bovine tuberculosis is zoonotic, meaning it can be transmitted from animals to humans. Humans can contract the disease by consuming unpasteurised dairy products or through direct contact with infected animals. National control programs aim to reduce the prevalence of bTB in cattle herds, thereby minimising the risk of transmission to humans and protecting public health. Bovine tuberculosis can have significant economic consequences for the livestock industry. Infected animals may experience reduced productivity, such as decreased milk production or weight loss, leading to financial losses for farmers [5,8]. Additionally, countries exporting livestock products must adhere to strict health and safety regulations, and the presence of bTB can result in trade restrictions and decreased market access. National control programs help to mitigate these economic impacts by controlling the spread of the disease and maintaining the health of cattle herds. Bovine tuberculosis can cause suffering and distress to infected animals. The disease can lead to respiratory problems, emaciation, and general debilitation, negatively impacting the welfare of affected cattle [56]. National control programs work to improve animal welfare by identifying and managing cases of bTB, thereby reducing the prevalence of the disease and minimising its impact on livestock [57]. Bovine tuberculosis can also affect wildlife populations, particularly in regions where wildlife and livestock interact. For example, in areas where bTB is endemic, wildlife species such as deer and badgers may serve as reservoirs for the disease, potentially transmitting it to cattle [58]. National control programs may include measures to monitor and manage bTB in wildlife populations, helping protect wildlife and livestock. Bovine tuberculosis is a global concern, and international cooperation is essential for effectively controlling its spread. National control programs often involve collaboration with neighbouring countries and participation in regional and international initiatives to combat the disease. By working together, countries can share knowledge and resources to implement more comprehensive control measures and prevent the spread of bTB across borders. National control programs for bTB play a critical role in safeguarding public health, supporting the livestock industry, promoting animal welfare, conserving wildlife, and facilitating international cooperation in disease control efforts [22,27].

In the present mapping study, it can be seen that Cundinamarca and other departments have been significantly affected by bTB. They should be the subject of further studies and activities to prevent and control it. Although only available for 2016, the population rate of bTB among animals was highest in Putumayo. As such, even more studies in this department should be carried out because there are no published studies about bTB or bovine diseases.

It is essential to acknowledge the multiple limitations of our study, as it being a descriptive retrospective one meant that we could not assess the specific associated factors related to the differences in bTB in Colombia. Underlying differences in farming in different parts of Colombia may account for the presentation of bTB breakdowns and the risk of infection. Also, the differences in dairy versus beef herds and in herds with a low turnover and few purchased animals versus herds made up of many purchased animals from multiple sources with a high turnover may impact the incidence of disease. As a retrospective study, multiple variables, including the number of veterinarians performing tests, were unavailable. Also, no other test results were available except those for tuberculin. However, tuberculin is an excellent tool for screening for bTB in multiple countries worldwide, and the goal is surveillance. This study was conducted precisely at the screening level. Nevertheless, we observed that no positive results were available for 2017 to 2019, and there was no information at all regarding testing in multiple departments in that period, which is another limitation, especially in terms of information availability if tests were performed in the country. For example, in 2019, only information about four departments was available. To speak of areas that are free of bTB, it is highly relevant to perform comprehensive assessments of the bovine populations, especially in previously endemic areas and in those considered at risk.

After this study, future studies in highly endemic areas of Colombia are expected. Despite its limitations, this study provides highly relevant information for veterinarians, physicians, and stakeholders in human and animal health in the country to better understand the epidemiology of bTB and its current implications. More robust studies should be developed in Colombia to contribute to a deeper understanding of this zoonotic disease that may even affect wildlife species [59,60,61].

To improve disease control, livestock movement should be considered; both within- and between-countries movement can contribute to the spread of bTB. Transmission can occur in regions with a significant cattle trade if infected animals are moved from one area to another without proper testing and quarantine measures [62,63,64].

Different countries in Latin America have implemented various control measures to combat bTB. These measures may include regular testing and surveillance of cattle herds, culling of infected animals, movement restrictions, and vaccination programs. However, the effectiveness of these measures can be influenced by factors such as funding availability, infrastructure, and stakeholder cooperation [65,66,67]. For example, in Brazil, control measures for bTB are primarily overseen and implemented by the Brazilian Ministry of Agriculture, Livestock, and Food Supply (MAPA) in collaboration with state-level veterinary services. Surveillance for bTB involves regular testing of cattle herds using tuberculin skin tests or other diagnostic methods to detect infected animals. This surveillance is usually conducted by veterinarians authorised by MAPA and local veterinary services. Additionally, monitoring herds with known bTB history or high-risk factors is crucial to prevent the spread of the disease [30,54,68].

Bovine tuberculosis can have substantial economic implications for the livestock industry in Latin America. Infected animals can suffer reduced productivity, and control measures can burden farmers and governments economically. Additionally, trade restrictions on livestock and livestock products due to bTB concerns can impact international trade relationships [8,69]. Bovine tuberculosis can have significant economic impacts on the agricultural sector, particularly in regions where it is prevalent. Infected cattle often suffer from reduced productivity, including decreased milk production, weight loss, and reduced fertility. In severe cases, infected animals may need to be culled, leading to direct economic losses for farmers [70,71].

Many countries implement trade restrictions on cattle and cattle products from regions affected by bTB. This can limit export opportunities for farmers and affect prices in domestic markets. Controlling the spread of bTB requires extensive testing, surveillance, and biosecurity measures, which can be costly for farmers, governments, and taxpayers. These costs can include testing kits, veterinarian services, and compensation for culled animals [72,73].

In areas where bTB is prevalent, restrictions on movement and activities related to cattle farming can also impact tourism and recreation industries. This is particularly relevant in regions where agriculture and rural tourism contribute significantly to the economy. Public awareness of bTB outbreaks can lead to reduced consumer confidence in beef and dairy products, affecting demand and prices in the market. Chronic bTB outbreaks can undermine the long-term viability of affected farms, leading to consolidation within the industry as smaller operations struggle to cope with the economic burden. Governments often allocate significant resources to combatting bTB through surveillance, vaccination programs, and compensation schemes. This diverts funds that could be allocated to other priorities, such as healthcare or education [74,75,76].

Agriculture is often the backbone of rural economies, and the economic impacts of bTB can ripple through these communities, affecting farmers, local businesses, and service providers. Bovine tuberculosis can have wide-ranging economic consequences, affecting livestock production, trade, government budgets, rural communities, and consumer confidence. Efforts to control and eradicate the disease are essential for animal health and welfare and for maintaining the economic viability of agricultural sectors worldwide [77,78].

A One Health approach, which involves collaboration between veterinary and medical professionals, is crucial for addressing the disease’s impact on animal and human health. Efforts to control bTB should include coordination among veterinary authorities, public health agencies, and other relevant stakeholders [51,79,80].

Bovine tuberculosis is primarily a cattle disease but can infect many mammalian species, including wildlife. In Colombia, wildlife hosts of bTB may include the white-tailed deer (*Odocoileus virginianus*), the wild boar (*Sus scrofa*), the mountain tapir (*Tapirus pinchaque*), the white-lipped peccary (*Tayassu pecari*), the collared peccary (*Pecari tajacu*), the spectacled bear (*Tremarctos ornatus*), the puma (*Puma concolor*), and the jaguar (*Panthera onca*); however, there is lack of studies on this. Livestock and wildlife interactions that could contribute to the spread of bTB often occur at the interface of human activities, such as agricultural expansion, livestock grazing, and habitat encroachment into wildlife areas, which occur in multiple rural areas of Colombia [81]. In other countries in Latin America, such as Argentina, Brazil, and Uruguay, this has been more comprehensively approached by researchers [81,82,83,84].

Due to its complex nature and multifaceted impacts, the One Health approach is crucial for addressing the control and elimination of bTB in low and middle-income countries (LMICs). Bovine tuberculosis is a zoonotic disease, meaning it can be transmitted from animals to humans, posing significant public health risks [85,86,87].

Bovine tuberculosis affects cattle and humans who come into contact with infected animals or their products. Additionally, the environment plays a role in the transmission dynamics of the disease. The One Health approach recognises these interconnected factors and emphasises collaboration across human, animal, and environmental health sectors to address bTB effectively [88,89]. This approach aims to prevent the zoonotic transmission of bTB by recognising and addressing the shared risks of disease transmission between animals and humans. This involves implementing improved surveillance, early detection, and control measures in animal and human populations [19,90]. This allows for the early detection of outbreaks and facilitates targeted interventions to prevent the disease’s further spread [91,92].

Bovine tuberculosis control and elimination require collaboration across multiple sectors, including agriculture, veterinary medicine, public health, and environmental management. The One Health approach promotes multisectoral collaboration to develop comprehensive strategies that address the root causes of bTB transmission and enhance disease control efforts [93,94].

Bovine tuberculosis can have significant economic impacts on livestock production and public health systems in LMICs. By adopting a One Health approach, interventions can be designed to control the spread of bTB and promote sustainable livelihoods for communities reliant on livestock farming [95,96].

Bovine tuberculosis is a global health issue with implications for food security, trade, and international public health. Implementing the One Health approach helps strengthen global health security by addressing emerging infectious diseases at their source and preventing their spread across borders [97].

Overall, the One Health approach recognises the interconnectedness of human, animal, and environmental health and emphasises collaborative efforts to address complex health challenges such as bTB in low and middle-income countries [98,99,100]. By integrating expertise from multiple disciplines and sectors, the One Health approach holds great promise for controlling and eventually eliminating bTB while safeguarding public health and promoting sustainable development.

## 5. Conclusions

Geographical information systems are essential for understanding the temporospatial behaviour of zoonotic diseases in Colombia, as is the case for bovine tuberculosis, with potential implications for Human and One Health. In this seminal approach, we were able to characterise this zoonotic disease in the country and provide maps to display the spatial distribution of bTB in the country’s departments for a significant period. Other studies on this disease, at different analytical levels, are expected.

## Figures and Tables

**Figure 1 vetsci-11-00220-f001:**
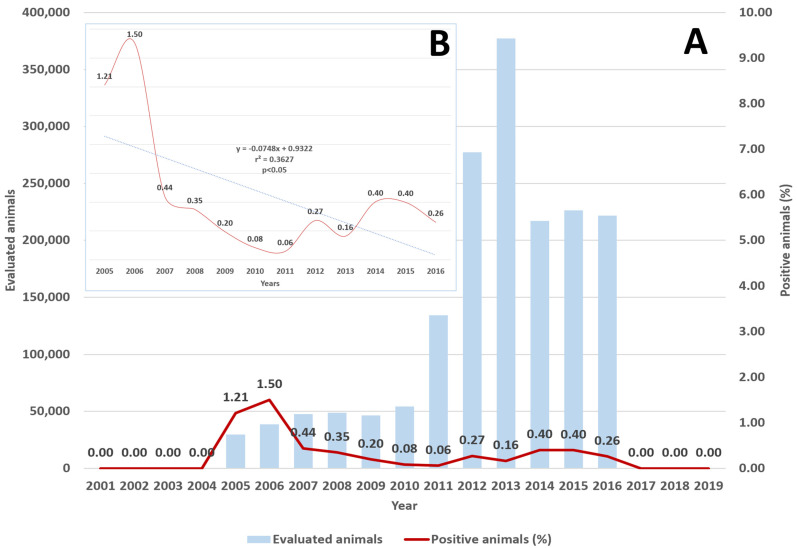
(**A**). Bovine tuberculosis, the proportion of positive animals per year (%), and number of studied animals, Colombia, 2001–2019. (**B**). Insert linear trends for 2005–2016, showing a statistically significant reduction (*p* < 0.05) during those years.

**Figure 2 vetsci-11-00220-f002:**
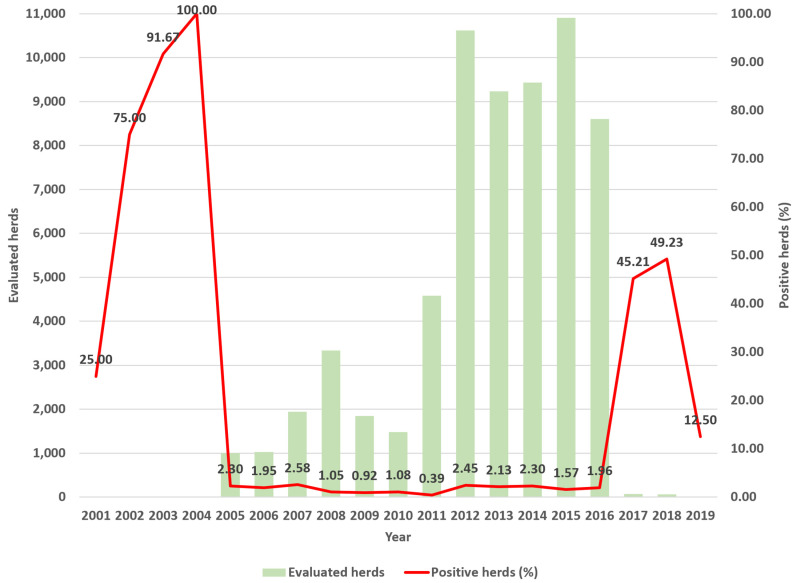
A. Bovine tuberculosis by herds, number of evaluated herds and proportion of positive herds per year (%), Colombia, 2001–2019.

**Figure 3 vetsci-11-00220-f003:**
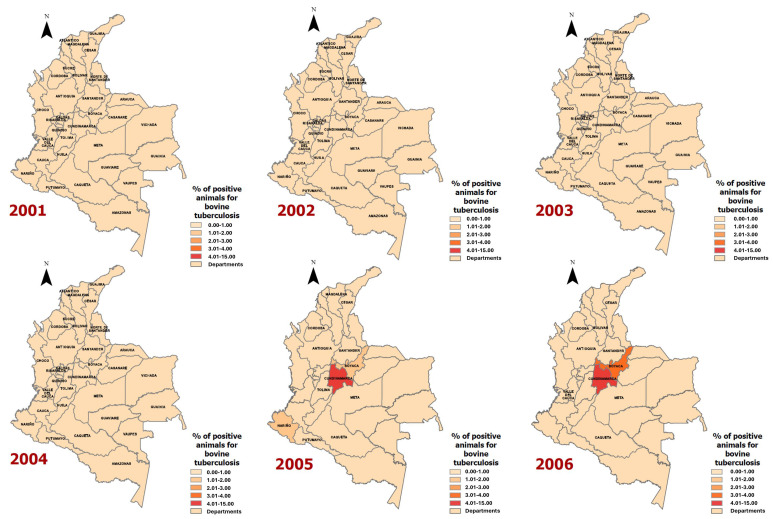
Proportion of animals positive for bovine tuberculosis in Colombia, 2001–2019.

**Figure 4 vetsci-11-00220-f004:**
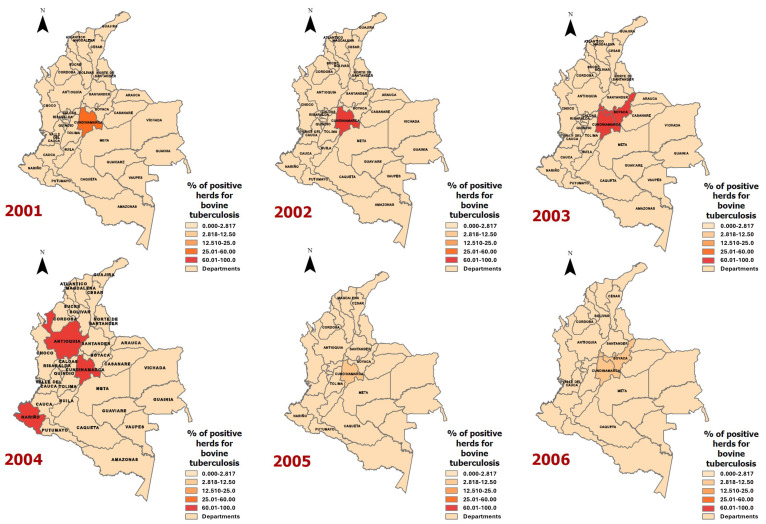
Proportion of herds positive for bovine tuberculosis in Colombia, 2001–2019.

**Figure 5 vetsci-11-00220-f005:**
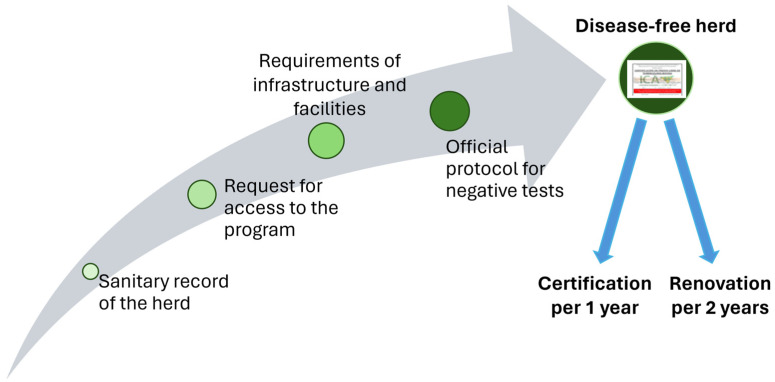
Process of certification of free-disease herds of the National Program for the Prevention, Control and Eradication of bTB.

**Table 1 vetsci-11-00220-t001:** Rates of bovine tuberculosis (cases/100,000 animals) per department in Colombia from 2016 to 2019.

Year	Department	Positive Animals	Number of Animals	Rate
2016	Putumayo	45	197,611	22.8
2016	Caldas	60	370,345	16.2
2016	Huila	53	415,246	12.8
2016	Cundinamarca	138	1,256,535	11
2016	Antioquia	221	2,632,125	8.4
2016	Santander	55	1,412,313	3.9
2016	Norte de Santander	3	389,694	0.8
2016	Arauca	3	1,048,543	0.3
2016	Bolivar	0	925,446	0
2016	Boyacá	0	748,701	0
2016	Caquetá	0	1,340,049	0
2016	Cauca	0	273,663	0
2016	Cesar	0	1,357,512	0
2016	Córdoba	0	1,942,770	0
2016	La Guajira	0	285,298	0
2016	Magdalena	0	1,207,764	0
2016	Meta	0	1,660,147	0
2016	Nariño	0	384,686	0
2016	Quindío	0	81,788	0
2016	Risaralda	0	109,117	0
2016	Sucre	0	862,008	0
2016	Tolima	0	547,647	0
2016	Valle	0	459,596	0
2016	Vaupés	0	1223	0
2016	Vichada	0	242,633	0
Total		578	20,152,460	76.1
2017	12 Departments Assessed	0	14,103,998	0
2018	9 Departments Assessed	0	25,044,896	0
2019	4 Departments Assessed	0	5,804,865	0

## Data Availability

Available upon reasonable request. Raw data is available at https://www.ica.gov.co/areas/pecuaria/servicios/epidemiologia-veterinaria/bol/epi.aspx?aliaspath=%2fAreas%2fPecuaria%2fServicios%2fEpidemiologia-Veterinaria%2fBol%2fEpi (accessed on 10 January 2022).

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
