# Peer review of "Mapping Bovine Tuberculosis in Colombia, 2001–2019"

_vetsci, 2024, doi:10.3390/vetsci11050220_

Round 1

Reviewer 1 Report

Comments and Suggestions for Authors

The authors describe the spatial and temporal distribution of bovine tuberculosis (bTB) in bovines in Colombia over a period of 18 years.

Overall, the paper is well written and the subject matter is original in the context of Colombia. The techniques applied are descriptive rather than analytical and are simple and robust. I have a number of comments and observations.

The background and overall description of bTB in terms of aetiology and impact are well addressed. In fact, there is probably too much detail considering the focus of this paper is on the mapping aspect of the disease in Colombia. The overall text could be shortened accordingly, especially in the discussion, but I would consider this a suggestion. However, what I found to be missing was a description of the Colombia bTB testing/surveillance program. I would consider this to be essential. There is no mention of the intensity of surveillance. Are all herds tested at fixed intervals? Are all regions tested at the same rate? Are herds re-tested if positive animals are found? Are they restricted in terms of animal movements for a period after a positive test is recorded? This context is important for the readers when appraising the ‘evaluated animals vs positive animals’ in the text.

Further, more detail on the actual test is required (Line 94-97: Where is the injection site (caudal or neck)? Is it a comparative test or a single injection? What are the cutoffs reaction (size, colour, etc.) for the animal to be classified as positive? Is there slaughterhouse surveillance? Do positive animals get flagged for post mortem examination? If lesions are found are they confirmed with bacteriology or culture? Just for example, in Table 1, Arauca 2016, there are 3 positive animals out of ~1 million. How many animals were tested? What is the sensitivity and specificity of the test? Even a very good test such as the SICCT (99.98% specificity) would give 200 false positives from a million test. (line 228 of discussion eludes to this but doesn’t explore Colombia’s testing regime). Adding the number of animals/herds tested to a table would give an indication of the uncertainty around the estimated annual incidence.

One of the aims of this study was to enable a better understanding the ‘behaviour’ of bTB in Colombia. The study itself simply describes the geographic incidence of bTB and doesn’t reflect on why these differences may occur. Are there underlying differences in husbandry in different parts of Colombia that may account for the presentation of bTB breakdowns and the risk of infection, e.g., Dairy versus beef herds (herds with low turnover and few purchased animals versus herds made up of many purchased animals from multiple sources with a high turnover).

Line 18-19: rewrite. ‘livestock countries’ not a good descriptor. The text in the introduction is clearer.

Line 63: ‘on the other hand’ isn’t really appropriate as these diseases readily infect multiple species and have similar zoonotic potential. Maybe re-emphasise that m.bovis is the dominant circulating strain in livestock in most countries.

Line 93: I’m not an immunologist, but BCG  is used solely as a vaccine and not as a diagnostic tuberculin.

Line 114: ‘as well as other…’

Line 125: ‘In the study of secondary sources, the main instrument’ suggest: ‘In the exploration of secondary data sources, the main resource…’

Line 126: ‘there are’ change to ‘these contain’

Figures 3,4,5 and 6 are all part of the same map series with the same legend intervals. A suggestion would be to merge these into one figure if possible. Additionally, it would be useful for the actual 0 (zero) values to be no colour (just outlined). The color ramp in the classifications isn’t ideal for such small images.

Line 206: herd ‘level’

No animal level data for 2016 and 2017? Should the animal level maps be excluded? I’m a bit confused about this statement and can’t find reference to it in the earlier text.

There is a considerable difference between animal and level incidence maps. Is this a function of these regions having fewer herds with larger herd sizes?

Line 282: Deer and badgers are recognised hosts of bTB in counties such as the UK and Ireland, what are the wildlife hosts in Colombia and where do the livestock/wildlife interactions occur most frequently – a brief evidenced description adding to line 306-307 would be useful for context.

Comments on the Quality of English Language

The quality of English was generally good. There was some repetition between the introduction and discussion and aspects of the discussion were a bit fragmented, with points relating to One Health overlapping with points made previously. The results sections was a bit hard to follow in terms of descriptions of regions but this is mainly down to my unfamiliarity of the geography of Columbia!

Author Response

Reviewer 1 Comments and Responses

The authors describe the spatial and temporal distribution of bovine tuberculosis (bTB) in bovines in Colombia over a period of 18 years.

Thanks a lot.

Overall, the paper is well written and the subject matter is original in the context of Colombia. The techniques applied are descriptive rather than analytical and are simple and robust. I have a number of comments and observations.

Thanks a lot for your comments. We appreciate your comments for the improvement of the article.

The background and overall description of bTB in terms of aetiology and impact are well addressed. In fact, there is probably too much detail considering the focus of this paper is on the mapping aspect of the disease in Colombia. The overall text could be shortened accordingly, especially in the discussion, but I would consider this a suggestion.

The original submitted text was shorter, but the journal asked to increase the text in general to at least 4,000 words.

However, what I found to be missing was a description of the Colombia bTB testing/surveillance program. I would consider this to be essential. There is no mention of the intensity of surveillance. Are all herds tested at fixed intervals? Are all regions tested at the same rate? Are herds re-tested if positive animals are found? Are they restricted in terms of animal movements for a period after a positive test is recorded? This context is important for the readers when appraising the ‘evaluated animals vs positive animals’ in the text.

Now, at Discussion we have detailed information regarding the bTB control program in Colombia. We have even included a Figure in order to explain the processes in the program. Regarding surveillance we detailed now more, how is performed, passive and active. Other aspects are also explained.

Further, more detail on the actual test is required (Line 94-97: Where is the injection site (caudal or neck)? Is it a comparative test or a single injection? What are the cutoffs reaction (size, colour, etc.) for the animal to be classified as positive? Is there slaughterhouse surveillance? Do positive animals get flagged for post mortem examination? If lesions are found are they confirmed with bacteriology or culture? Just for example, in Table 1, Arauca 2016, there are 3 positive animals out of ~1 million. How many animals were tested? What is the sensitivity and specificity of the test? Even a very good test such as the SICCT (99.98% specificity) would give 200 false positives from a million test. (line 228 of discussion eludes to this but doesn’t explore Colombia’s testing regime). Adding the number of animals/herds tested to a table would give an indication of the uncertainty around the estimated annual incidence.

This part focus more on the detail of prospective studies. It is important to remember that our study is a retrospective approach, based on secondary data. That label of detail on each animal is not available. Other details, regarding slaughter of positive animals are also included now.

One of the aims of this study was to enable a better understanding the ‘behaviour’ of bTB in Colombia. The study itself simply describes the geographic incidence of bTB and doesn’t reflect on why these differences may occur. Are there underlying differences in husbandry in different parts of Colombia that may account for the presentation of bTB breakdowns and the risk of infection, e.g., Dairy versus beef herds (herds with low turnover and few purchased animals versus herds made up of many purchased animals from multiple sources with a high turnover).

As you stated, our study it is descriptive. This initial observational study is not analytical not focused on associated factors. However, we included that as limitation of the current study.

Line 18-19: rewrite. ‘livestock countries’ not a good descriptor. The text in the introduction is clearer.

Agree, we modified it.

Line 63: ‘on the other hand’ isn’t really appropriate as these diseases readily infect multiple species and have similar zoonotic potential. Maybe re-emphasise that m.bovis is the dominant circulating strain in livestock in most countries.

Done. Included.

Line 93: I’m not an immunologist, but BCG  is used solely as a vaccine and not as a diagnostic tuberculin.

Agree. Deleted

Line 114: ‘as well as other…’

Done. Corrected.

Line 125: ‘In the study of secondary sources, the main instrument’ suggest: ‘In the exploration of secondary data sources, the main resource…’

Done. Modified.

Line 126: ‘there are’ change to ‘these contain’

Done. Changed.

Figures 3,4,5 and 6 are all part of the same map series with the same legend intervals. A suggestion would be to merge these into one figure if possible. Additionally, it would be useful for the actual 0 (zero) values to be no colour (just outlined). The color ramp in the classifications isn’t ideal for such small images.

We developed then separately, for an ease understanding, and for the format of the journal articles.

Line 206: herd ‘level’

Done. Included.

No animal level data for 2016 and 2017? Should the animal level maps be excluded? I’m a bit confused about this statement and can’t find reference to it in the earlier text.

Unfortunately, detailed data for 2016 and 2017 was not available. We developed maps based on the available data.

There is a considerable difference between animal and level incidence maps. Is this a function of these regions having fewer herds with larger herd sizes?

This is a natural difference observed in multiple studies that assess both levels. We have observed that in other studies for other diseases in bovines.

Line 282: Deer and badgers are recognised hosts of bTB in counties such as the UK and Ireland, what are the wildlife hosts in Colombia and where do the livestock/wildlife interactions occur most frequently – a brief evidenced description adding to line 306-307 would be useful for context.

Now, we have included a text regarding that information, at the Discussion. We included some additional relevant references about that.

Reviewer 2 Report

Comments and Suggestions for Authors

Authors performed a descriptive, retrospective study about tuberculosis in Colombia. The manuscript is interesting and give important information for the Academia and the country. However, in my humble opinion, these outputs are mainly the ones that public resources are giving. Therefore, authors have to give a differential point of view. 

Some other issues: 

- no simple summary

- ln 30. GIS: what is it?

- Introduction: too long sentences without references. Please provide more references to support the arguments you are giving.

- ln 76. testing? There are some ways to test. Describe briefly because it can be different from Europe to America, for example, or even different within the country. I would include the following paragrpah in here.

- Introduction: what is about transmission thorugh wild animals?

- M&M: this section needs to be improved substantially. As it is a retrospective study, we need to know exactly how they got data and compared it. For example: number of veterinarians performing tests? When do they find it as positive? Do you have any other result appart tuberculin? 

I think, too many issues missing in the experimental design...

- Results: period 2001-2005: are there results or no tuberculin was performed? In figure 1 no animals were positive, but in figure 2 a lot of herds were positive. How can this be supported? It does happen the same between 2017-2019.

- Results: are all the sections in the country tested? Because it seemed to be mainly distributed in the central region and left side.

- Table 1: just showing some of the departments, but previously you indicated that census were about 27million. Here you have no more than 8m. What is happening with the rest? Was not it tested? Are there no results?

All these issues have been discussed appropriately, however discussion is too long. I recommend to shorten. 

Comments on the Quality of English Language

minor

Author Response

Reviewer 2 Comments and Responses

Authors performed a descriptive, retrospective study about tuberculosis in Colombia. The manuscript is interesting and give important information for the Academia and the country. However, in my humble opinion, these outputs are mainly the ones that public resources are giving. Therefore, authors have to give a differential point of view.

Thanks for your comments. It is important to clarify that such “outputs” are NOT available publicly, we used raw data, we analyzed them, and we developed the presented GIS-based maps.

Some other issues:

- no simple summary

Now, a Simple Summary has been included.

- ln 30. GIS: what is it?

Now, described.

- Introduction: too long sentences without references. Please provide more references to support the arguments you are giving.

Done. We have now included the corresponding references. The article now has 13 additional references, for a total of 102.

- ln 76. testing? There are some ways to test. Describe briefly because it can be different from Europe to America, for example, or even different within the country. I would include the following paragrpah in here.

Done. Now included.

- Introduction: what is about transmission thorugh wild animals?

We included now a significant paragraph about it at the Discussion. As, also requested by other reviewer.

- M&M: this section needs to be improved substantially. As it is a retrospective study, we need to know exactly how they got data and compared it. For example: number of veterinarians performing tests? When do they find it as positive? Do you have any other result appart tuberculin?

It is important to remember that our study is a retrospective approach, based on secondary data. And that multiple aspects are not available, including some of those mentioned. However, considering that, we have include that as a limitation. There no other tests available apart from tuberculin.

I think, too many issues missing in the experimental design...

This is NOT an experimental study. It is an observational, descriptive study.

- Results: period 2001-2005: are there results or no tuberculin was performed? In figure 1 no animals were positive, but in figure 2 a lot of herds were positive. How can this be supported? It does happen the same between 2017-2019.

The availability of data was limited. We struggle with the available data. It is important to remember that we have data at two analysis level: individual animals, and herds. Then, results, are different. E.g. if in a herd, one animal is positive, the herd is considered positive. If there are nine additional herds, the prevalence at herd levels is 1/10, 10%, but if those herds in total have 1000 animals, and no other animal is positive, that would be 1/1000, 0,1% prevalence at individual level.

- Results: are all the sections in the country tested? Because it seemed to be mainly distributed in the central region and left side.

Testing is performed in all the departments of Colombia.

- Table 1: just showing some of the departments, but previously you indicated that census were about 27million. Here you have no more than 8m. What is happening with the rest? Was not it tested? Are there no results?

There were no available tests for the rest.

All these issues have been discussed appropriately, however discussion is too long. I recommend to shorten.

The original submitted text was shorter, but the journal asked to increase the text in general to at least 4,000 words.

Reviewer 3 Report

Comments and Suggestions for Authors

The manuscript titled " Mapping Bovine Tuberculosis in Colombia, 2010-2019" provides an overview of bovine tuberculosis in Colombia. The authors provide temporal-spatial distribution of cases using a retrospective cross-sectional descriptive study. Overall the manuscript is well written and organized. The introduction clearly states the important of this study, figures are clear and the discussion is detailed. I only have one minor comment to share. Figures 3-9 have very small font for the departments. Its hard to read. But the color schemes do allow to see which departments have positive cases.

Author Response

Reviewer 3 Comments and Responses

The manuscript titled " Mapping Bovine Tuberculosis in Colombia, 2010-2019" provides an overview of bovine tuberculosis in Colombia. The authors provide temporal-spatial distribution of cases using a retrospective cross-sectional descriptive study. Overall the manuscript is well written and organized. The introduction clearly states the important of this study, figures are clear and the discussion is detailed.

Thanks a lot for all your positive comments.

I only have one minor comment to share. Figures 3-9 have very small font for the departments. Its hard to read. But the color schemes do allow to see which departments have positive cases.

We have now increase the size of the figures.

Round 2

Reviewer 2 Report

Comments and Suggestions for Authors

Authors answered all my comments. However, there are important lacks that, in my humble opinion, makes this study not strong enough because of missing data/variables:

- M&M: this section needs to be improved substantially. As it is a retrospective study, we need to know exactly how they got data and compared it. For example: number of veterinarians performing tests? When do they find it as positive? Do you have any other result appart tuberculin?

It is important to remember that our study is a retrospective approach, based on secondary data. And that multiple aspects are not available, including some of those mentioned. However, considering that, we have include that as a limitation. There no other tests available apart from tuberculin.

IMPORTANT LIMITATION 

I think, too many issues missing in the experimental design...

This is NOT an experimental study. It is an observational, descriptive study.

I obviously know, however, retrospective studies do need an experimental design described in M&M. Experimental design is the way to denominate how the study was performed

- Results: period 2001-2005: are there results or no tuberculin was performed? In figure 1 no animals were positive, but in figure 2 a lot of herds were positive. How can this be supported? It does happen the same between 2017-2019.

The availability of data was limited. We struggle with the available data. It is important to remember that we have data at two analysis level: individual animals, and herds. Then, results, are different. E.g. if in a herd, one animal is positive, the herd is considered positive. If there are nine additional herds, the prevalence at herd levels is 1/10, 10%, but if those herds in total have 1000 animals, and no other animal is positive, that would be 1/1000, 0,1% prevalence at individual level.

Lacking of all data availability difficults the impact of the results of the study and make them less strong...

- Table 1: just showing some of the departments, but previously you indicated that census were about 27million. Here you have no more than 8m. What is happening with the rest? Was not it tested? Are there no results?

There were no available tests for the rest.

Same concern about data availability that commented previously.

Comments on the Quality of English Language

minor

Author Response

Reviewer 2 Comments and Answers

Authors answered all my comments. However, there are important lacks that, in my humble opinion, makes this study not strong enough because of missing data/variables:

This is a descriptive retrospective study that pretends to characterise the geographical distribution of bovine tuberculosis in Colombia, a neglected topic in the country and Latin America. Similar studies are also lacking. Our study has limitations, but beyond that, it provides valuable information for the community regarding the distribution not available elsewhere. We developed original maps for bovine tuberculosis in Colombia that had never existed before.

 - M&M: this section needs to be improved substantially. As it is a retrospective study, we need to know exactly how they got data and compared it. For example: number of veterinarians performing tests? When do they find it as positive? Do you have any other result appart tuberculin?

It is important to remember that our study is a retrospective approach, based on secondary data. And that multiple aspects are not available, including some of those mentioned. However, considering that, we have include that as a limitation. There no other tests available apart from tuberculin.

IMPORTANT LIMITATION 

In retrospective studies, you will always not have all the variables you want. We would like to have many more variables. However, this is a retrospective, even ecological study from bovine populations, not individual subjects. Our limitations have been acknowledged in the Discussion. To solve that, prospective studies are needed. We have commented on that in our Discussion now and are extending it. We explain more now regarding the collection of data and processing in Methods.

I think, too many issues missing in the experimental design...

This is NOT an experimental study. It is an observational, descriptive study.

I obviously know, however, retrospective studies do need an experimental design described in M&M. Experimental design is the way to denominate how the study was performed

We have now provided more details regarding the design of this study in Materials and Methods.

- Results: period 2001-2005: are there results or no tuberculin was performed? In figure 1 no animals were positive, but in figure 2 a lot of herds were positive. How can this be supported? It does happen the same between 2017-2019.

The availability of data was limited. We struggle with the available data. It is important to remember that we have data at two analysis level: individual animals, and herds. Then, results, are different. E.g. if in a herd, one animal is positive, the herd is considered positive. If there are nine additional herds, the prevalence at herd levels is 1/10, 10%, but if those herds in total have 1000 animals, and no other animal is positive, that would be 1/1000, 0,1% prevalence at individual level.

Lacking of all data availability difficults the impact of the results of the study and make them less strong...

Regarding data, certainly, there is “some” data lacking. However, if you see the whole study, we have a lot of data and a lot of maps, enabling us to develop valuable maps in a significant number of years to show the geographical distribution of bovine tuberculosis in Colombia. In the Discussion, we have stated the limitations of this study. We hope the reviewer checks the whole document and gives the precise value of this contribution to the epidemiology of bovine tuberculosis in a country lacking these studies.

- Table 1: just showing some of the departments, but previously you indicated that census were about 27million. Here you have no more than 8m. What is happening with the rest? Was not it tested? Are there no results?

There were no available tests for the rest. 

Same concern about data availability that commented previously.

There was a mistake in the previous answer. We have the data of the census and the estimates for the rest of the animal population for 2016, 2017, 2018 and 2019, but there were no positive results. We have modified the table now, including that. In those years, there were departments with no information regarding test availability. However, the table is now more complete and comprehensive. The census for 2019 was 27m, but unfortunately, it is obvious that not all animals are tested, which it should be, but precisely that is the situation and are calling on that now also at the Discussion.

Round 3

Reviewer 2 Report

Comments and Suggestions for Authors

Authors accomplished some of my issues. With the new provided version, which is substantially improved, I already understood the importance of this data to be shared with the Academia. Especially, with the Colombian bovine practitioners, researchers, farmers... Therefore, I suggest some other issues to improve the manuscript prior accepting for publication. 

1) Avoid using "introduction", "objective", etc. in the abstract. Improve readability to make it more attractive. Readers go ahead to read the whole manuscript depending on how they feel attracted by the abstract.

2) I still emphasizing on the importance of showing the knowledge gap in the introduction. I would encourage authors to reduce introduction and include the impact of sharing these data with the Academia.

3) ln 167. ICA DATABASES. Please provide more information, maybe even a link or something to double check.

4) figure 1: improve quality. Statistical reduction in panel b: explain when... 

5) figure 2: p values?

6) figures 3 to 6: merge them all into one figure of one page. As you are using red colours it can be quickly identified and at one sight you can see it all. Interesting to have it all together.

7) figures 7 to 10: same comment than before.

8) table 1 provide interesting information by department. I feel is very long... Think about reduction: there are a lot of 0 cases in some regions.

9) Discussion is extremely long, please shorten it.

Comments on the Quality of English Language

minor

Author Response

Reviewer 2:

Authors accomplished some of my issues. With the new provided version, which is substantially improved, I already understood the importance of this data to be shared with the Academia. Especially, with the Colombian bovine practitioners, researchers, farmers... Therefore, I suggest some other issues to improve the manuscript prior accepting for publication.

Thanks a lot for your comments. We really appreciate them.

1) Avoid using "introduction", "objective", etc. in the abstract. Improve readability to make it more attractive. Readers go ahead to read the whole manuscript depending on how they feel attracted by the abstract.

“Introduction”, deleted. “Objective” too. We have edited the abstract, improving it and making it more attractive. Thanks for your recommendations.

2) I still emphasizing on the importance of showing the knowledge gap in the introduction. I would encourage authors to reduce introduction and include the impact of sharing these data with the Academia.

Now we have include more at Introduction regard the knowledge gap and the importance of the study. We have also reduced the Introduction from 963 to 867 words.

3) ln 167. ICA DATABASES. Please provide more information, maybe even a link or something to double check.

Done. The link to ICA has been now included. Also, at the end of the article (Data Availability).

4) figure 1: improve quality. Statistical reduction in panel b: explain when...

We modified the Figure, and we extended the explanation regarding the reduction.

5) figure 2: p values?

Figure 2 has no p-values. This figure has no significance tests, so it does not apply in this case.

6) figures 3 to 6: merge them all into one figure of one page. As you are using red colours it can be quickly identified and at one sight you can see it all. Interesting to have it all together.

Done. We have now merged Figures 3 to 6, into new Figure 3.

7) figures 7 to 10: same comment than before.

Done. We have now merged Figures 7 to 10, into new Figure 4.

8) table 1 provide interesting information by department. I feel is very long... Think about reduction: there are a lot of 0 cases in some regions.

Now, we have edited the Table to reduce it, considering the comment.

9) Discussion is extremely long, please shorten it.

Done. We have shortened from 2,770 words to 2,515 words. It is important to mention that there is also a limit in the reductions, as the journal clearly asked us to have at least 4,000 words. The original articles are requested to have at least 4,000 words. Now, we have 4,481.
